# Shading Stress at Different Grain Filling Stages Affects Dry Matter and Nitrogen Accumulation and Remobilization in Fresh Waxy Maize

**DOI:** 10.3390/plants12091742

**Published:** 2023-04-23

**Authors:** Haohan Sun, Wei Li, Yuwen Liang, Guanghao Li

**Affiliations:** 1Jiangsu Key Laboratory of Crop Genetics and Physiology, Yangzhou University, Yangzhou 225009, China; 2Jiangsu Key Laboratory of Crop Cultivation and Physiology, Yangzhou University, Yangzhou 225009, China; 3Jiangsu Co-Innovation Center for Modern Production Technology of Grain Crops, Yangzhou University, Yangzhou 225009, China

**Keywords:** dry matter, fresh waxy maize, N remobilization, shading stress, yield

## Abstract

Shading stress caused by plum rain season, which overlapped with grain filling process of fresh waxy maize in Southern China, significantly affected crop productivity. In order to investigate the effects of shading at different stages after pollination on the yield, accumulation, and remobilization of dry matter and nitrogen (N) in fresh waxy maize, field experiments were conducted, including shading at 1–7 (Z1), 8–14 (Z2), 15–21 (Z3), and 1–21 (Z4) days after pollination in 2020 and 2021. The results showed that shading reduced the fresh ear and grain yield and increased moisture content in Suyunuo5 (SYN5) and Jingkenuo2000 (JKN2000) compared to natural lighting treatment (CK). The ear yield decrease was more severe in Z4 (43.5%), followed by Z1 (29.7%). Post-silking dry matter and N accumulation and remobilization were decreased under shading stress, and those were lowest in Z4, followed by Z1. The remobilization of pre-silking dry matter and N were increased by shading stress, and the increase was highest in Z4, followed by Z1. The harvest index of dry matter and N was lowest in Z4 and second-lowest in Z1. In conclusion, shading decreased yield by affecting accumulation and remobilization of post-silking dry matter and N, and the impact was more serious when it introduced early during grain filling stage in fresh waxy maize production.

## 1. Introduction

In Southeast China, overcast weather frequently occurs from mid-June to mid-July and causes insufficient light during the growth periods, which has become one of the main factors restricting maize yield, increasing the risk of global food production and nutrition security under the background of global climatic change. The main material of crops comes from photosynthesis. Maize is a typical C_4_ crop, and sufficient light is the necessary condition to ensure its high and stable yield [1]. In recent years, due to climate change and other factors, the average sunshine hours during maize growth period have decreased by 13%, and the average solar radiation has decreased by 4% [2]. The lack of sunlight will become an important limiting factor for the further improvement of maize yield in the future [3,4]. Previous studies have shown that the effect of shading at post-silking on maize yield was greater than shading at pre-silking [5,6,7]. Shading stress at the grain filling stage caused poor pollination, insufficient photosynthetic products and delayed maturity [8,9]. Some studies also showed that different degrees of shading after pollination also led to a reduction in starch accumulation, reducing maize yield and degrading starch quality [10,11].

Under shading stress, the light energy utilization efficiency of C_4_ plants decreased, and the overall photosynthetic rate of leaves decreased [12,13]. Shading stress led to the separation of the plasma membrane of mesophyll cells at post-silking, the destruction of the inner membrane system, and the structural damage of mesophyll cells [14]. Shading stress reduced the activity of photosynthesis-related enzymes [15], thus reducing the net photosynthetic rate of leaves and leading to the decline of photosynthetic production capacity [16]. Leaf photosynthesis is the main source of grain filling [17], and the declined leaf photosynthetic capacity under shading restricted grain development and filling requirements, which promoted the mobilization of water-soluble carbohydrate stored in stem to the developing grains, but it still cannot make up for the loss of dry matter production after pollination under insufficient light [8,18]. However, the effects of shading and excision of different parts of durum wheat confirmed the high contribution of both carbons assimilated by ears and remobilized from stems to grain filling and the relatively low contribution of leaves to grain filling [19]. Under shading conditions, due to insufficient dry matter supply, root development was significantly hindered, resulting in a decline in root activity and a reduction in nitrogen, phosphorus, and potassium absorption, affecting the growth and development of the plants [20]. The development of female and male panicles and the process of pollination were significantly affected under shading stress, resulting in a significant reduction in grain number [21,22]. The development of grain is regulated by endogenous hormones. Under shading stress, the sucrose supplied to maize grains decreased; the content of abscisic acid (ABA) in grains increased; and the content of indole acetic acid (IAA), gibberellin (GA), and zeatin riboside (ZR) decreased [23], resulting in a significant decrease in the number and volume of endosperm cells in grains. Additionally, the duration of the filling peak, active filling period, and maximum and average filling rate were decreased [24]; the grain plumpness was insufficient; and the grain weight was significantly reduced [7]. In summary, shading stress reduced the photosynthesis of maize, resulting in a lack of energy and material supply in a series of subsequent physiological processes, and significantly reduced the yield. However, previous studies have mainly focused on the dry matter and N accumulation and remobilization in plants that were harvested at maturity, and limited information is available about crops that are harvested at the milk stage.

The grain filling stage is the key period for the division and differentiation of endosperm cells and the formation of grain yield, which is extremely sensitive to adverse conditions [25]. Shading is sink-limited; previous studies have identified the effects of different degrees of shading stress on the yield of waxy maize at the filling stage [26]. The effect of shading on the development of endosperm was different at different growth stages after pollination. Shading stress at 1–15 days after pollination reduced the number of endosperm cells, resulting in reduction of starch granule number and significantly delayed the development of starch granules; shading at 16–30 days after pollination led to smaller endosperm cell volume and poorer filling state, resulting in significantly lower granules grain volume [7]. The development of fresh waxy maize is of great significance to the adjustment of China’s planting industry [27], and due to the influence of the plum rain climate in the Yangtze River basin, maize sown in spring often suffers from insufficient light stress during the post-silking growth periods. Considering that the fresh waxy maize plants are harvested at the mid-grain-filling (milk) stage [28], the accumulation and remobilization of dry matter and N may be different from those in cereal crops that are harvested at maturity. Grains accumulate carbon and nitrogen provided by different organ sources during the whole life of the plants [29]. Thus, they represent the ultimate integration of different C and N supplies. Understanding the accumulation and remobilization of dry matter and N in different organs under shading conditions at different stages after pollination could provide reference for avoiding or alleviating the influence of low sunlight during grain filling in fresh waxy maize production.

## 2. Results

### 2.1. Yield

Shading stress at different stages after pollination had a significant impact on the fresh ear yield, grain yield and moisture content of the two varieties (Figure 1). Shading reduced the fresh ear and grain yield compared with CK in two years. The average yield was lowest in Z4, followed by Z1. Compared to CK, the ear yields of Z1, Z2, Z3, and Z4 were decreased by 28.7%, 21.8%, 14.7%, and 39.6%, respectively, in SYN5 and by 30.5%, 22.8%, 25.1%, and 46.7%, respectively, in JKN2000. The grain yields of Z1, Z2, Z3, and Z4 were decreased by 31.9%, 23.6%, 16.0%, and 43.7%, respectively, in SYN5 and by 32.1%, 21.7%, 24.6%, and 44.7%, respectively, in JKN2000. The average fresh ear and grain yields of JKN2000 were significantly higher than SYN5, and the decrease in JKN2000 was more severe under shading stress. Shading at different stages increased the moisture content in grain to different extents. The increase was highest at Z4 (4.1% in SYN5 and 4.2% in JKN2000), followed by Z1 (2.7% in both varieties).

### 2.2. Dry Matter and Nitrogen Accumulation at Post-Silking

Shading stress at different stages after pollination significantly reduced the accumulation of dry matter and N at post-silking in two years (Figure 2). The average accumulation of dry matter and N at post-silking was lowest in Z4, followed by Z1. Compared to CK, the dry matter accumulations of Z1, Z2, Z3, and Z4 at post-silking were decreased by 30.3%, 15.4%, 16.4%, and 67.6%, respectively, in SYN5 and by 29.8%, 14.0%, 18.2%, and 56.7%, respectively, in JKN2000. The N accumulations of Z1, Z2, Z3, and Z4 at post-silking were decreased by 26.9%, 15.1%, 17.8%, and 63.6%, respectively, in SYN5 and by 28.5%, 12.4%, 19.5%, and 56.5%, respectively, in JKN2000.

### 2.3. Post-Silking Remobilization of Dry Matter and Nitrogen

Shading stress at different stages after pollination significantly reduced the post-silking remobilization of dry matter and N in two years (Figure 3). The average post-silking remobilization of dry matter and N was lowest in Z4, followed by Z1. Compared to CK, the post-silking remobilization of dry matter in Z1, Z2, Z3, and Z4 were decreased by 33.7%, 17.6%, 17.5%, and 75.3%, respectively, in SYN5 and by 34.1%, 10.1%, 14.6%, and 58.9%, respectively, in JKN2000. The post-silking N remobilization of Z1, Z2, Z3, and Z4 were decreased by 27.5%, 16.5%, 17.5%, and 66.1%, respectively, in SYN5 and by 30.0%, 9.9%, 18.0%, and 57.7%, respectively in JKN2000. The average post-silking remobilization of dry matter and N in SYN5 was significantly higher than in JKN2000. Similar trends were found in 2020 and 2021.

### 2.4. Pre-Silking Remobilization of Dry Matter and Nitrogen

Year, variety, shading stress, and their interactions had significant effects on the pre-silking remobilization of dry matter and N in two years (Table 1 and Table 2). The average remobilization of pre-silking dry matter increased under shading stress. The remobilization amount from pre-silking leaf (stem) to grain (REP) and remobilization efficiency (REE) from stem and leaf were highest in Z4, followed by Z1. Similar trends were found in 2020 and 2021. In SYN5, stem REPs of Z1, Z2, and Z4 were positive in 2020; in 2021, this was only positive of Z4. All the leaf REPs were negative. In JKN2000, all the stem REPs were positive in 2020; only Z3 and Z4 were in 2021. The increase in total dry matter REP was highest in Z4, followed by Z1 in SYN5 and JKN2000. Those of Z2 and Z3 were similar in SYN5, but that of Z3 was significantly higher than Z2 in JKN2000.

Shading stress increased the average N remobilization at pre-silking. In SYN5, the N REP and REE from stem and leaf were highest in Z4, followed by Z1. In JKN2000, N REPs from stem were highest in Z4 and CK, N REPs from leaf were highest in Z4 and Z1, and total N REP and REE were highest in Z4, followed by CK, Z1, and Z3 (had no significant difference); that in Z2 was lowest. In SYN5, only the N REP from stem and total of Z4 were positive in 2020. In JKN2000, all the N REPs from stem were positive. Two varieties had inconsistent responses to shading stress at different stages after pollination.

### 2.5. Harvest Index of Dry Matter and Nitrogen

Shading stress at different stages after pollination had a significant impact on the harvest index (HI) of dry matter and N (Figure 4). The average HI was lowest in Z4, followed by Z1. Compared with CK, the dry matter HI of Z4 was decreased by 5.7% in SYN5, and those of Z1, Z2, and Z3 were similar. In JKN2000, the dry matter HIs of Z1, Z2, and Z4 were decreased by 5.1%, 1.9%, and 5.3%, respectively, and that of Z3 was similar to CK. The N HIs of Z1, Z2, Z3, and Z4 were decreased by 4.2%, 2.9%, 3.2%, and 8.6%, respectively, in SYN5 and by 6.6%, 5.5%, 3.3%, and 9.5%, respectively, in JKN2000. The average HI of dry matter and N in JKN2000 was similar to SYN5, and the decrease in JKN2000 was more severe under shading stress.

## 3. Discussion

The plant total dry matter and yield primarily come from direct deposition of post-silking photo-assimilate and remobilization of the non-structural carbohydrate reserved from vegetative organs at pre-silking [30,31]. The carbon resource of crop is mainly produced via leaf photosynthesis at post-silking, but the photosynthetic production capacity of leaves decreased significantly under insufficient light conditions [8,16] because of mesophyll cell structural damage and activity reduction of photosynthetic enzymes [32], which decreased maize yield harvested at both the milk [33] and maturity stages [34]. This is consistent with the results of the present study, which suggest that shading at different grain filling stages decreased ear and grain yield by reducing post-silking accumulation of dry matter and decreasing the remobilization from stem and leaf. This study also demonstrated that the yield decrease was more serious when shading was introduced early (1–7 days after pollination) during the grain filling stage in fresh waxy maize production, possibly because this stage is the key stage to grain formation and because shading limited the size of the sink, which reduced grains per ear at the early stage of grain filling and reduced grain weight at later stage [35]. A study also demonstrated that shading stress remarkably reduced the endosperm filling status, and the impact degree was the largest in the early stage of grain filling [7]. Though the pre-silking REP and REE of dry matter from stem and leaf were increased under shading conditions, the increase was not enough to make up for the decrease of photosynthetic products at post-silking, ultimately resulting in yield lost. The HI was unaffected under three short-term shading stresses (Z1, Z2, and Z3) in SYN5 but significantly decreased in JKN2000 due to the differences in pre-silking remobilization from stem and leaf.

High accumulation of dry matter and N are the target traits in crops to improve grain yield [36]. Previous studies have demonstrated that shading decreased the plants’ nutrient absorption by hindering root development [20] and the soil nutrient cycling process [37,38]. The present study showed that shading at different grain filling stages decreased post-silking N accumulation and decreased the N remobilization from stem and leaf. This finding was consistent with previous studies, which showed that the activities of nitrate reductase, glutamine synthetase, and glutamate synthase that are involved in N metabolism were downregulated by shading stresses in maize harvested at maturity [39]. In wheat [40], cotton [41], and tomatoes [42], the research results also showed that shading stress decreased the activities of enzymes related to N metabolism. This study also demonstrated the post-silking N amount decrease was more serious when shading was introduced early after pollination in fresh waxy maize, which had similar trends with the post-silking N remobilization. At the milk stage, N REP from stem and leaf at pre-silking had significant differences between SYN5 and JYN2000. The N remobilization from stem occurred at the milk stage in JKN2000 but not in SYN5, except Z4. This is also the main reason for the different responses of the two varieties in NHI to shading at different grain filling stages. The N content and activity of related enzymes are closely associated with plant photosynthetic capacity and leaf senescence [43]. This can also explain the consistent change in dry matter and N accumulation and remobilization post-silking. This study clarified that the negative impact was more serious when insufficient light was introduced early during the grain filling stage in fresh waxy maize production. On this basis, we will start with cultivation measures in the future to seek methods such as spraying exogenous growth regulators and topdressing N fertilizer to alleviate the shading stress on yield loss, and we will provide scientific basis for the realization of stress resistance, stable yield, and high efficiency in fresh waxy maize production.

## 4. Materials and Methods

### 4.1. Experimental Design

We conducted a field trial in 2020 and 2021 at the Yangzhou University farm (32°30′ N, 119°25′ E) in Jiangsu Province, China. Daily precipitation, average air temperature, and sunlight hours during maize growing period are shown in Figure 5.

Treatments were arranged in a split plot design. The main plot was assigned with two fresh waxy maize varieties, namely Suyunuo5 (SYN5, used in the national fresh waxy maize regional test as the control variety) and Jingkenuo2000 (JKN2000, having the largest planted area of waxy maize in China). SYN5 was provided by the Yanjiang Institute of Agricultural Sciences in Jiangsu, and JKN2000 was provided by the Beijing Academy of Agriculture and Forestry Sciences. The subplot included four shading treatments: shading at 1–7 days after pollination (Z1), shading at 8–14 days after pollination (Z2), shading at 15–21 days after pollination (Z3), and shading at 1–21 days after pollination (Z4). Natural light treatment was taken as the control (CK). After pollination, black shading net with 50% shading degree was used to build a shed. The distance between the shading net and the maize canopy was always 2–2.5 m to ensure that the field microclimate under the shading shed was basically consistent with the natural lighting conditions in the field. Light intensity and photosynthetically active radiation under different treatments were determined using quantum light meters (3415FQF, Spectrum technologies, Inc., Aurora, IL, USA) in 2021 (Table 3). The sowing dates were March 20 and March 24 in 2020 and 2021, respectively, and pollination dates were June 10 and June 14, respectively. Two varieties were harvested at the milk stage (the 23rd day after pollination) on July 2 and July 6 in 2020 and 2021, respectively. Maize was double-row planted (0.8 and 0.4 cm) according to local traditional method. Each plot was 72 m^2^ (10 m × 7.2 m), with 60,000 plants ha^−1^. Slow-released compound fertilizer (N/P_2_O_5_/K_2_O = 27%/9%/9%) was applied at sowing time with N/P_2_O_5_/K_2_O rates of 225/75/75 kg ha^−1^, respectively. The depth of furrow fertilization was 10 cm. Other field management followed farmers’ conventional practices.

### 4.2. Sampling and Measurements

Three representative plants of two varieties were collected and separated into leaves and stems (including sheaths and tassels) at the silking stage, and into leaves, stems, cobs, and grains at the milk stage. The samples were dried in a forced air oven at 105 °C for half an hour and then dried at 80 °C to a constant weight and weighed separately. After weighing, the samples were ground using a cyclone sample mill with a fine mesh (0.5 mm). The N concentrations of different organs were determined using the micro-Kjeldahl method. The following parameters referring to dry matter and N were calculated as previously described [27,44].

Plant dry matter (nitrogen) accumulation amount = Plant N concentration × Plant dry weight;

Post-silking dry matter (nitrogen) accumulation = dry matter (nitrogen) accumulation at the milk stage − dry matter (nitrogen) accumulation at the silking stage;

Dry matter (nitrogen) remobilization efficiency (%, REE) = 100 × dry matter (nitrogen) remobilization from pre-silking leaf (stem) to grain/dry matter (nitrogen) accumulation in leaf (stem) at the silking stage;

Remobilization amount of dry matter (nitrogen) of post-silking vegetative organ photosynthate = grain dry weight at the milk stage − dry matter (nitrogen) remobilization from pre-silking vegetative organs to grain;

Harvest index (HI, %) = 100 × grain dry weight/total dry matter at the milk stage;

Nitrogen HI (NHI, %) = 100 × grain N accumulation at the milk stage/total N accumulation at the milk stage

At the milk stage, 30 representative ears were collected continuously from each plot. Based on the average ear weight, three uniform ears were selected to determine the fresh grain yield from each treatment.

### 4.3. Statistical Analysis

All data were subjected to ANOVA in the General Linear Model module of SPSS. Comparisons among treatments were based on Duncan’s test at the 0.05 probability level (*p* < 0.05). Figures were generated with the Sigma Plot 12.0 program.

## 5. Conclusions

Shading stress at different stages after pollination decreased the post-silking dry matter and nitrogen accumulation and remobilization, increased the remobilization of pre-silking stem and leaf dry matter, and decreased the harvest index of dry matter and N in both fresh waxy maize varieties. When shading was introduced early during grain filling stage (1–7 days after pollination), the decrease in yield and accumulation and remobilization of dry matter and N was more serious. Therefore, we recommend that measures such as spraying exogenous growth regulators and topdressing N fertilizer to alleviate the insufficient light should be carried out before silking.

## Figures and Tables

**Figure 1 plants-12-01742-f001:**
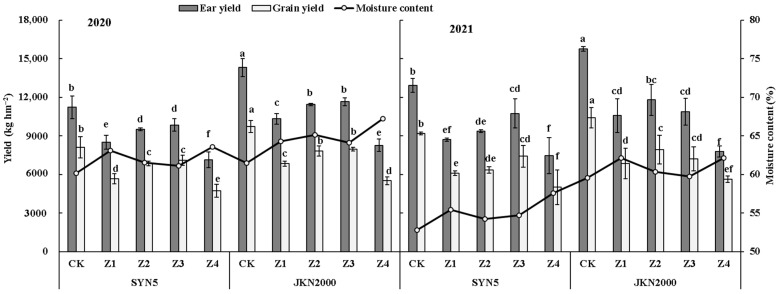
Effects of shading stress at different grain filling stages on yield of fresh waxy maize. SYN5, Suyunuo5; JKN2000, Jingkenuo2000; CK, natural light; Z1, Z2, Z3, and Z4, shading at 1–7, 8–14, 15–21, and 1–21 days, respectively, after pollination; different letters above the bars in each year were significantly different at a 5% probability level.

**Figure 2 plants-12-01742-f002:**
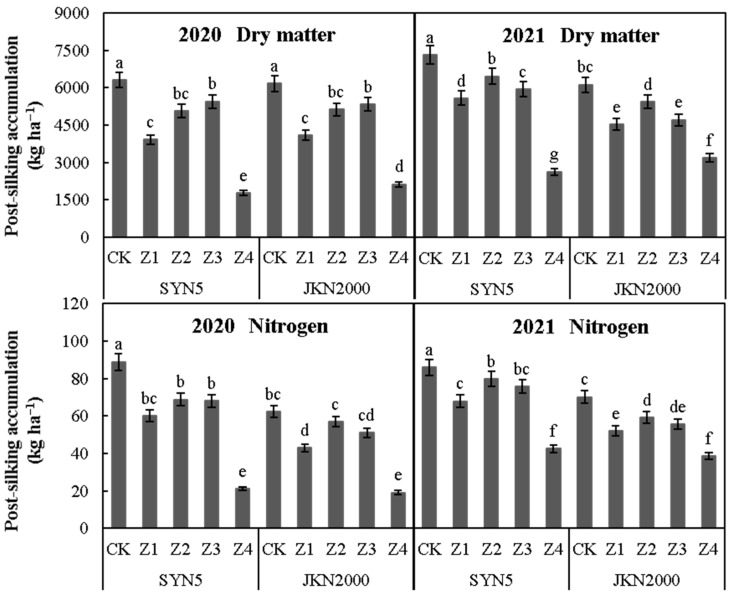
Effects of shading stress at different grain filling stages on the post-silking accumulation of dry matter and nitrogen in fresh waxy maize. SYN5, Suyunuo5; JKN2000, Jingkenuo2000; CK, natural light; Z1, Z2, Z3, and Z4, shading at 1–7, 8–14, 15–21, and 1–21 days, respectively, after pollination; different letters above the bars in each year were significantly different at a 5% probability level.

**Figure 3 plants-12-01742-f003:**
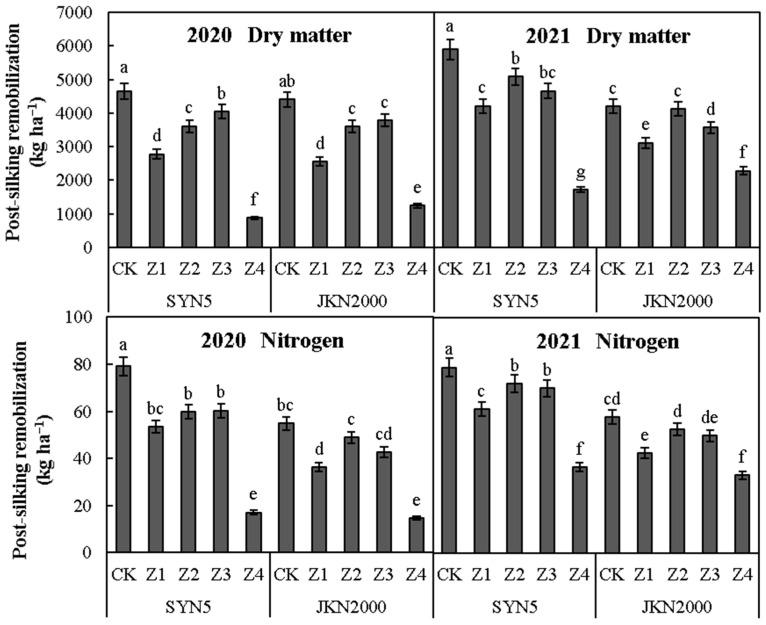
Effects of shading stress at different grain filling stages on the post-silking remobilization of dry matter and nitrogen in fresh waxy maize. SYN5, Suyunuo5; JKN2000, Jingkenuo2000; CK, natural light; Z1, Z2, Z3, and Z4, shading at 1–7, 8–14, 15–21, and 1–21 days, respectively, after pollination; different letters above the bars in each year were significantly different at a 5% probability level.

**Figure 4 plants-12-01742-f004:**
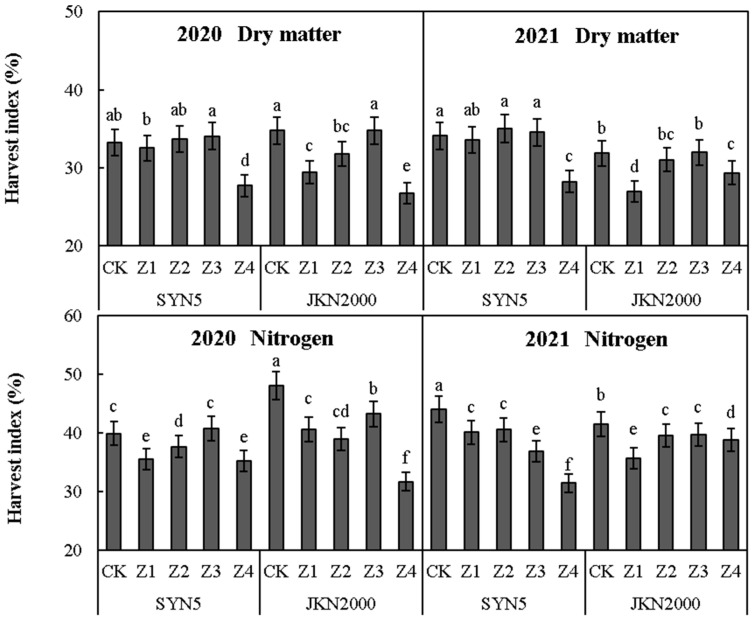
Effects of shading stress at different grain filling stages on harvest index of fresh waxy maize. SYN5, Suyunuo5; JKN2000, Jingkenuo2000; CK, natural light; Z1, Z2, Z3, and Z4, shading at 1–7, 8–14, 15–21, and 1–21 days, respectively, after pollination; different letters above the bars in each year were significantly different at a 5% probability level.

**Figure 5 plants-12-01742-f005:**
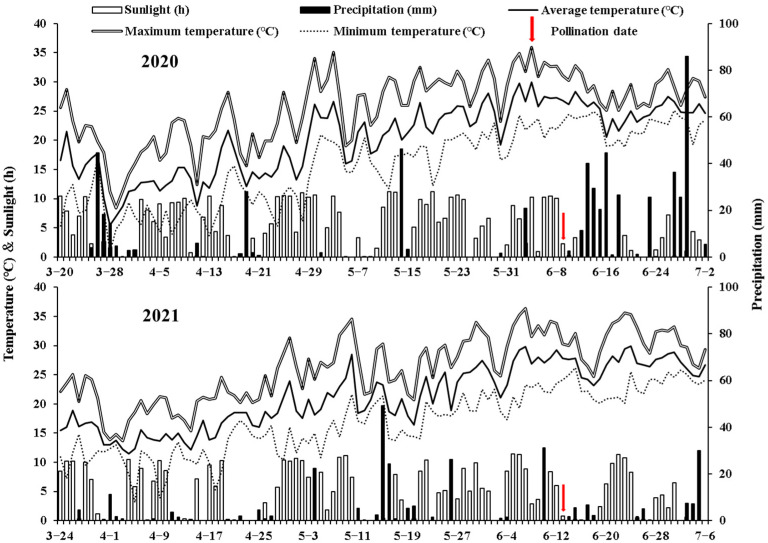
Daily precipitation, average temperature, and sunlight during maize growth seasons in 2020 and 2021.

**Table 1 plants-12-01742-t001:** Effects of shading stress at different grain filling stages on the remobilization of pre-silking dry matter in vegetative organs.

Year	Variety	Shading Treatment	Stem		Leaf		Total	
			REP (kg ha^−1^)	REE (%)	REP (kg ha^−1^)	REE (%)	REP (kg ha^−1^)	REE (%)
2020	SYN5	CK	−458.8 h	−13.6 f	−591.6 h	−49.8 f	−1050.4 j	−23.1 f
		Z1	332.8 c	9.9 c	−359.0 f	−30.2 d	−26.2 f	−0.6 c
		Z2	106.2 d	3.2 d	−481.2 g	−40.5 e	−375.0 h	−8.2 d
		Z3	−38.6 g	−1.1 e	−601.5 h	−50.6 f	−640.1 i	−14.0 e
		Z4	1149.4 a	34.1 a	−274.8 e	−23.1 c	874.6 b	19.2 a
	JKN2000	CK	4.6 f	0.1 e	−103.6 d	−4.9 b	−99.0 g	−1.6 c
		Z1	372.2 c	9.0 c	102.4 a	4.9 a	474.6 c	7.6 b
		Z2	72.8 e	1.8 de	−76.2 c	−3.6 b	−3.4 e	−0.1 c
		Z3	332.4 c	8.1 c	−95.6 d	−4.5 b	236.8 d	3.8bc
		Z4	962.2 b	23.4 b	20.6 b	1.0 ab	982.8 a	15.8 a
2021	SYN5	CK	−1435.6 h	−52.0 e	−535.8 g	−38.0 de	−1971.4 j	−47.3 f
		Z1	−564.6 f	−20.5 d	−377.4 e	−26.7 cd	−942.0 g	−22.6 d
		Z2	−697.2 g	−25.3 d	−660.2 h	−46.8 e	−1357.4 i	−32.6 e
		Z3	−705.0 g	−25.6 d	−458.2 f	−32.5 d	−1163.2 h	−27.9 de
		Z4	378.2 b	13.7 a	−174.8 b	−12.4 b	203.4 b	4.9ab
	JKN2000	CK	−46.4 d	−1.2 bc	−275.8 d	−13.3 b	−322.2 e	−5.3 bc
		Z1	−26.2 d	−0.7 bc	−238.6 c	−11.5 b	−264.8 d	−4.4 bc
		Z2	−139.0 e	−3.5 c	−423.2 f	−20.4 c	−562.2 f	−9.3 c
		Z3	123.0 c	3.1 b	−252.4 cd	−12.2 b	−129.4 c	−2.1 b
		Z4	495.2 a	12.5 a	−67.6 a	−3.3 a	427.6 a	7.1 a
ANOVA								
Year (Y)			301.2 **	357.2 **	307.7 **	20.5 **	115.6 **	106.4 **
Variety (V)			162.7 **	205.1 **	296.8 **	191.9 **	145.3 **	137.1 **
Shading (S)			222.5 **	478.4 **	674.2 **	147.5 **	374.3 **	282.9 **
Y × V			84.8 **	149.3 **	47.4 **	28.2 **	68.4 **	184.3 **
Y × S			25.7 **	40.5 **	165.4 **	27.4 **	77.2 **	45.0 **
V × S			40.3 **	127.4 **	54.7 **	31.7 **	53.8 **	68.7 **
Y × V × S			29.4 **	167.1 **	12.8 **	39.5 **	40.6 **	49.9 **

REP, remobilization amount from pre-silking leaf (stem) to grain; REE, remobilization efficiency; SYN5, Suyunuo5; JKN2000, Jingkenuo2000; CK, natural light; Z1, Z2, Z3, and Z4, shading at 1–7, 8–14, 15–21, and 1–21 days, respectively, after pollination; mean values within column in each year followed by different letters are significantly different at a 0.05 probability level. ** represent significant at the 0.01 probability levels.

**Table 2 plants-12-01742-t002:** Effects of shading stress at different grain filling stages on the remobilization of pre-silking nitrogen in vegetative organs.

Year	Variety	Shading Treatment	Stem		Leaf		Total	
			REP (kg ha^−1^)	REE (%)	REP (kg ha^−1^)	REE (%)	REP (kg ha^−1^)	REE (%)
2020	SYN5	CK	−17.1 f	−72.1 g	−8.3 bc	−37.5 f	−25.4 e	−55.5 f
		Z1	−8.0 e	−33.5 f	−8.1 bc	−36.7 f	−16.0 d	−35.0 e
		Z2	−8.7 e	−36.5 f	−8.1 bc	−37.0 f	−16.8 d	−36.7 e
		Z3	−4.3 d	−18.2 e	−9.7 c	−44.2 g	−14.1 d	−30.7 d
		Z4	11.7 ab	49.1 a	−5.2 b	−23.6 e	6.5 b	14.1 ab
	JKN2000	CK	11.3 ab	39.0 b	−3.7 b	−9.5 c	7.7 b	11.2 b
		Z1	5.8 c	19.9 d	2.8 a	7.1 a	8.6 b	12.6 b
		Z2	5.3 c	18.3 d	−5.6 b	−14.4 d	−0.3 c	−0.5 c
		Z3	8.4 b	28.9 c	0.4 ab	1.1 b	8.8 b	13.0 b
		Z4	13.5 a	46.5 ab	−0.6 ab	−1.5 b	12.9 a	19.0 a
2021	SYN5	CK	−13.1 de	−63.7 f	−7.1 b	−26.9 d	−20.2 e	−43.0 f
		Z1	−7.6 cd	−37.1 d	−7.3 b	−27.7 d	−15.0 d	−31.8 e
		Z2	−10.0 d	−48.6 e	−10.5 c	−39.8 e	−20.5 e	−43.7 f
		Z3	−14.8 e	−71.8 g	−10.0 c	−37.7 e	−24.8 f	−52.6 g
		Z4	−4.9 c	−23.6 c	−3.4 ab	−13.0 b	−8.3 c	−17.7 d
	JKN2000	CK	7.7 a	26.1 a	−7.9 b	−20.3 c	−0.2 b	−0.3 b
		Z1	5.3 b	17.8 b	−4.7 ab	−12.0 b	0.6 b	0.9 b
		Z2	5.9 ab	20.1 ab	−7.9 b	−20.4 c	−2.0 bc	−2.9 c
		Z3	7.0 a	23.7 a	−7.5 b	−19.3 c	−0.5 b	−0.8 b
		Z4	7.5 a	25.4 a	1.0 a	2.5 a	8.5 a	12.4 a
ANOVA								
Year (Y)			61.9 **	66.6 **	261.8 **	7.9 *	120.1 **	100.7 **
Variety (V)			117.9 **	83.3 **	127.9 **	138.7 **	153.9 **	171.6 **
Shading (S)			213.8 **	180.9 **	239.0 **	237.4 **	415.1 **	349.5 **
Y × V			8.1 *	19.3 **	30.9 **	196.7 **	2.6	0.1
Y × S			84.9 **	61.8 **	48.3 **	33.4 **	48.4 **	49.5 **
V × S			130.9 **	134.9 **	52.3 **	46.7 **	97.3 **	121.4 **
Y × V × S			45.3 **	35.3 **	42.9 **	34.0 **	65.3 **	64.0 **

REP, remobilization amount from pre-silking leaf (stem) to grain; REE, remobilization efficiency; SYN5, Suyunuo5; JKN2000, Jingkenuo2000; CK, natural light; Z1, Z2, Z3, and Z4, shading at 1–7, 8–14, 15–21, and 1–21 days, respectively, after pollination; mean values within column in each year followed by different letters are significantly different at a 0.05 probability level. * and ** represent significant at the 0.05 and 0.01 probability levels.

**Table 3 plants-12-01742-t003:** Light intensity and photosynthetically active radiation under different shading treatments (2021).

Date (Month/Date)	Light Intensity (Lux)					Photosynthetically Active Radiation (μmol m^−2^ s^−1^)				
	CK	Z1	Z2	Z3	Z4	CK	Z1	Z2	Z3	Z4
6.9	100,800	49,236	101,100	100,456	49,736	1355	670	1286	1397	580
6.10	85,312	48,536	84,312	82,312	49,577	1059	486	1100	1065	512
6.11	84,352	54,562	83,352	82,352	53,562	1137	523	1037	1237	557
6.12	123,008	66,637	122,008	121,008	65,637	1897	845	1765	1812	893
6.13	92,148	60,054	91,148	93,148	59,054	1391	712	1223	1308	693
6.14	63,008	29,617	62,008	64,008	27,617	906	357	880	912	445
6.15	121,512	57,617	120,512	122,512	56,617	1900	865	1800	185	905
6.16	67,637	66,637	21,562	65,637	24,562	945	913	435	925	392
6.17	66,066	65,066	30,916	64,066	33,916	1020	920	535	1107	525
6.18	76,719	75,719	41,047	74,719	42,912	1213	1113	639	1013	607
6.19	73,706	72,706	34,772	71,706	33,701	1186	1086	506	1114	570
6.20	49,636	48,636	19,899	47,636	19,074	776	758	341	726	328
6.21	60,181	59,181	21,621	58,181	20,621	955	946	396	925	402
6.22	81,378	80,378	42,374	79,378	43,558	1297	1182	587	1306	640
6.23	2582	2475	2260	1323	1681	35	36	38	7	6
6.24	83,099	82,099	81,099	42,299	41,015	1315	1150	1320	620	632
6.25	84,713	83,713	82,713	41,976	40,763	1275	1391	1296	621	585
6.26	88,878	89,878	87,878	43,020	48,005	1390	1253	1274	611	578
6.27	39,629	38,629	40,629	17,887	18,608	687	703	673	272	295
6.28	90,492	89,492	88,492	36,431	38,794	1597	1471	1538	714	740
6.29	99,271	98,271	97,271	38,970	37,579	1478	1434	1326	660	671

## Data Availability

Informed consent was obtained from all subjects involved in the study. All the data and code used in this study can be requested by email to the corresponding author Guanghao Li at guanghaoli@yzu.edu.cn.

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
