# Peer review of "Shading Stress at Different Grain Filling Stages Affects Dry Matter and Nitrogen Accumulation and Remobilization in Fresh Waxy Maize"

_plants, 2023, doi:10.3390/plants12091742_

Round 1

Reviewer 1 Report

Methodology with little detail. Absence of information on experimental design and number of repetitions. Absence of how each variable was calculated. I understand that 80C for drying the material is inadequate, with possible losses of N. 

Material and methods item is presented after results and discussion, which is inappropriate. Few variables form measurements and only two growth cycles were evaluated.

Reviewer 2 Report

 General comment:

Insufficient amount of light is one of the most important factors influencing negatively crop production, especially in such kind of plants like maize. The main question of the research was: how shading conditions at different stages after pollination in fresh waxy maize production affect the accumulation and remobilization of dry matter and N. The practical aspect was to provide reference for avoiding or alleviating the influence of low sunlight during grain filling. According to the literature the topic is original in terms of maize production. The new element in subject area compared with other published material is that the authors investigated crops that are harvested at the milk stage, while  previous studies have mainly focused on the dry matter and N accumulation and remobilization in plant that were harvested at full maturity. The results which showed differences between varieties are especially valuable – they may give basis to alleviation the negative effects in crop production by using particular varieties of maize.

I do find this work interesting and valuable. However, it needs a bit of correction. Statistics needs improvement or explanations.

Detailed comments:

I have a question about statistics: why in Figures 1 - 4 you have chosen to analyze both hybrids together, while in Tables 1 and 2 you performed analysis separately for each hybrid? Please at least explain.

Material and methods

Lines 252 – 253 – I think brief description how all these parameters  were calculated will be beneficial, so a reader will not have to look at another articles.

Line 168 – 169 “Shading reduced the HI of dry matter and N compared with CK in two years” – this sentence is not accurate. Looking at Figure 4 there were incidents when values in shaded treatments were even a bit higher than in control – for example Z2 and Z3 in SYN5 in 2021 Dry matter (although not statistically significant)

Conclusions

Lines 269 – 270 – “the measures alleviated the insufficient light…”  could you suggest here what kind of measures?

Reviewer 3 Report

These are my main comments on the manuscript entitled “Shading stress at different grain filling stages affects dry matter and nitrogen accumulation and remobilization in fresh waxy maize.” The authors examined the impacts of shading on two different maize hybrids at different grain-filling stages. They concluded that shading stress decreased the post-silking dry mat and nitrogen accumulation and remobilization, increased the remobilization of pre-silking stem and leaf dry matter, and decreased the harvest index of dry matter and N in both maize hybrids.

Overall the manuscript is clear, organized, and well-structured. The introduction was well-written and gave a good background of the main idea of the study.

The results are well-designed and statistically analyzed.

Abstract:

The abstract does not contain the goal or the hypothesis of the study. Please define it.

Line 14: What does the number mean? (1-7, 8-14, 15-21, etc.) Are they days?

Keywords:

Please arrange the keywords in alphabetical order.

Introduction:

Line 60: Define each abbreviation when they are first used.

Materials and Methods:

Line 247: Please correct: Three representative plants of…

Line 249: Use superscript for 23rd

Reviewer 4 Report

The paper evaluated the accumulation and remobilization of dry matter and N under shading conditions at different stages after pollination. It is important for fresh waxy maize production under the new situation of expanding special crops in China. The manuscript is written well. The authors should pay attention to the following questions.

1. Please describe the differences between the four shading treatments in the Abstract section.

2. Why did this experiment carry out four stages of shading treatment during grain filling stage?

3. What changes in the climate have caused insufficient light?

4. Please harmonize the usage of ‘variety’ and ‘hybrid’ in the manuscript.

5. Line 139 Please add the full names of ‘REP’ and ‘REE’ in the manuscript.

6. Please introduce the source of seeds in the Materials and Methods.

7. Please correct ‘shading’ to ‘treatment’ in Table 1 and Table 2.

8. Please unify the fonts in all figures.

9. Please describe the definition of the red arrow in the legend of Figure 5.

Reviewer 5 Report

The manuscript deals with the impact of shading stress at different seed filling stages on accumulation of nitrogen and dry matter.

Authors have used  a black shading net to disturb the light reception by Maize plants. They have harvested plants at differents grain filling stages and assessed dry matter and N in several plant parts. Authors havce used two hybrids in this study. The used experimental design is right.

The manuscript is interesting, even not original, and brings new information.

1- The most important shortcoming in this manuscript is the lack of literature background. Many studies have been done on other poaceae. see some of these references

Hannachi, L.; Deléens, E.; Gate, P. Nitrogen and carbon isotope composition of wheat grain: Alteration due to sink-source modifications at flowering. Rapid Commun. Mass Spectrom. 1996, 19, 979–986.

Araus, J.L.; Santiveri, P.; Bosch-Serra, D.; Royo, C.; Romagosa, I. Carbon isotope ratios in ear parts of triticale.
Plant Physiol. 1992, 100, 1033–1035.

Araus, J.L.; Febrero, A.; Vendrell, P. Epidermal conductance in different parts of durum wheat grown under
Mediterranean conditions. The role of epicuticular waxes and stomata. Plant Cell Environ. 1991, 14, 545–558.

Maydup, M.L.; Antonietta, M.; Graciano, C.; Guiamet, J.J.; Tambussi, E.A. The contribution of the awns of
bread wheat (Triticum aestivum L.) to grain filling: Responses to water deficit and the effects of awns on ear
temperature and hydraulic conductance. Field Crops Res. 2014, 167, 102–111.

https://doi.org/10.11118/actaun202068060937

Arous et al. Bulgarian Journal of Agricultural Science, 26 (No 4) 2020, 809–815

doi:10.3390/ijms19010056

https://doi.org/10.1111/jac.12109

This fact limits authors to develop a mechnistic discussion of their results. What a pity !!

2- has a more accurate light measurement been made?

3- the interactions between the studied factors are neither displayed nor discussed

minor remark

L235 please modify split ploy design by split plot design

Round 2

Reviewer 5 Report

AUthors have considered carefully all recommendations and their manuscript has been imrpoved. Thank you for considering positively the remarks done on this manuscript